# FSNet: Compression of Deep Convolutional Neural Networks by Filter Summary

**Yingzhen Yang**[1], **Jiahui Yu**[2], **Nebojsa Jojic**[3], **Jun Huan**[4], **Thomas S. Huang**[2]

[1] School of Computing, Informatics, and Decision Systems Engineering, Arizona State University
`yingzhen.yang@asu.edu`
[2] Beckman Institute, University of Illinois at Urbana-Champaign
`{jyu79,t-huang1}@illinois.edu`
[3] Microsoft Research
`jojic@microsoft.com`
[4] StylingAI Inc.
`lukehuan@shenshangtech.com`

## Abstract

We present a novel method of compression of deep Convolutional Neural Networks (CNNs) by weight sharing through a new representation of convolutional filters. The proposed method reduces the number of parameters of each convolutional layer by learning a 1D vector termed Filter Summary (FS). The convolutional filters are located in FS as overlapping 1D segments, and nearby filters in FS share weights in their overlapping regions in a natural way. The resultant neural network based on such weight sharing scheme, termed Filter Summary CNNs or FSNet, has a FS in each convolution layer instead of a set of independent filters in the conventional convolution layer. FSNet has the same architecture as that of the baseline CNN to be compressed, and each convolution layer of FSNet has the same number of filters from FS as that of the basline CNN in the forward process. With compelling computational acceleration ratio, the parameter space of FSNet is much smaller than that of the baseline CNN. In addition, FSNet is quantization friendly. FSNet with weight quantization leads to even higher compression ratio without noticeable performance loss. We further propose Differentiable FSNet where the way filters share weights is learned in a differentiable and end-to-end manner. Experiments demonstrate the effectiveness of FSNet in compression of CNNs for computer vision tasks including image classification and object detection, and the effectiveness of DFSNet is evidenced by the task of Neural Architecture Search.

## 1 Introduction

Deep Convolutional Neural Networks (CNNs) have achieved stunning success in various machine learning and pattern recognition tasks by learning highly semantic and discriminative representation of data (LeCun et al., 2015). Albeit the power of CNNs, they are usually over-parameterized and of large parameter space, which makes it difficult for deployment of CNNs on mobile platforms or other platforms with limited storage. Moreover, the large parameter space of CNNs encourages researchers to study regularization methods that prevent overfitting (Srivastava et al., 2014).

In the recently emerging architecture such as Residual Network (He et al., 2016) and Densely Connected Network (Huang et al., 2017), most parameters concentrate on convolution filters, which are used to learn deformation invariant features in the input volume. The deep learning community has developed several compression methods of reducing the parameter space of filters and the entire neural network, such as pruning (Luo et al., 2017; Li et al., 2017; Anwar et al., 2017), weight sharing and quantization (Han et al., 2016; Tung & Mori, 2018; Park et al., 2017) and low-rank and sparse representation of the filters (Ioannou et al., 2016; Yu et al., 2017). Quantization based methods (Tung & Mori, 2018; Park et al., 2017) may not substantially improve the execution time of CNNs, and many methods (Lebedev & Lempitsky, 2016; Zhang et al., 2018) have also been proposed to improve the inference speed of CNNs.

Weight sharing has been proved to be an effective way of reducing the parameter space of CNNs. The success of deep compression (Han et al., 2016) and filter pruning (Luo et al., 2017) suggest that there is considerable redundancy in the parameter space of filters of CNNs. Based on this observation, our goal of compression can be achieved by encouraging filters to share weights.

In this paper, we propose a novel representation of filters, termed Filter Summary (FS), which enforces weight sharing across filters so as to achieve model compression. FS is a 1D vector from which filters are extracted as overlapping 1D segments. Each filter in the form of a 1D segment can be viewed as an "unwrapped" version of the conventional 3D filter. Because of weight sharing across nearby filters that overlap each other, the parameter space of convolution layer with FS is much smaller than its counterpart in conventional CNNs. In contrast, the model compression literature broadly adopts a two-step approach: learning a large CNN first, then compressing the model by various model compression techniques such as pruning, quantization and coding (Han et al., 2016; Luo et al., 2017), or low-rank and sparse representation of filters (Ioannou et al., 2016; Yu et al., 2017).

Our FSNet is novel with the following contributions:

- We propose Filter Summary (FS) as a compact 1D representation for convolution filters. Filters in Filter Summary are 1D segments, as opposed to the conventional representation of filters as 3D arrays. FS is quantization friendly so as to achieve significant compression ratio.

- We propose a fast convolution algorithm for convolution with FS, named Fast Convolution by Filter Summary (FCFS), taking advantage of the 1D representation of filters in FS. Our FCFS is significantly different from the integral image method for fast convolution in Weight Sampling Network (WSNet) (Jin et al., 2018). The integral image method for WSNet is primarily designed for 1D CNNs where filters are intrinsically 1-dimensional and 1D convolution is performed. Such method cannot be directly extended to handling the convolution in regular 2D CNNs where filters are 3-dimensional in an efficient manner. Although one can approximate higher-dimensional convolution by 1D convolution, the approximation error is inevitable. By representing filters as overlapping 1D segments in 1D space, FCFS performs exact convolution with compelling computational acceleration ratio, which is the product of the compression ratio for convolution and the first spatial size of filter.

- In order to learn an optimal way filters share weights, we propose Differentiable Filter Summary Network, DFSNet. DFSNet represents the location of filters in each FS as learnable and differentiable parameters, so that the location of filters can be learned and adjusted accordingly. In this way, potential better location of filters can be obtained by training DFSNet in an end-to-end manner so as to guarantee the competitive accuracy of DFSNet.

### 1.1 RELATED WORKS

The idea of using overlapping patterns for compact representation of images or videos is presented in Epitome (Jojic et al., 2003), which is developed for learning a condensed version of Gaussian Mixture Models (GMMs). FSNet is inspired by Epitome and it uses overlapping structure to represent filters. Weight Sampling Network (WSNet) (Jin et al., 2018) also studies overlapping filters for compression of 1D CNNs. However, WSNet is not primarily designed for regular 2D CNNs, and its fast convolution method cannot be straightforwardly extended to the case of convolution in regular 2D CNNs without approximation.

### 1.2 NOTATIONS

Throughout this paper, we use $m : n$ to indicate integers between $m$ and $n$ inclusively, and $[n]$ is defined as $1 : n$. We use subscripts to indicate the index of an element of a vector, and $\mathbf{v}_{m:n}$ indicates a vector consisting of elements of $\mathbf{v}$ with indices in $m : n$.

## 2 FORMULATION

We propose Filter Summary Convolutional Neural Networks (FSNet) in this section. The Filter Summary (FS) is firstly introduced, and then the fast convolution algorithm named Fast Convolution by Filter Summary (FCFS) is developed. We then introduce the training of FSNet, FSNet with Weight Quantization (FSNet-WQ) and Differentiable FSNet (DFSNet).

## 2.1 Filter Summary (FS)

We aim to reduce the parameter space of convolution layers, or convolution filters. We propose Filter Summary (FS) as a compact representation of the filters. The filters are overlapping 1D segments residing in the Filter Summary. Each filter in a FS is a 1D segment comprised of consecutive elements of the FS. It is widely understood that a regular convolution filter is a 3D array with two spatial dimensions and one channel dimension. In FS, each segment representing a filter can be viewed as the "unwrapped" version of the corresponding regular 3D filter by concatenating all the elements of the 3D filter into a vector in channel-major order. In the sequel, filter has a 1D representation as a segment in FS without confusion. Figure 1(a) shows an example of a FS. Figure 1(b) shows how filters are located in the FS.

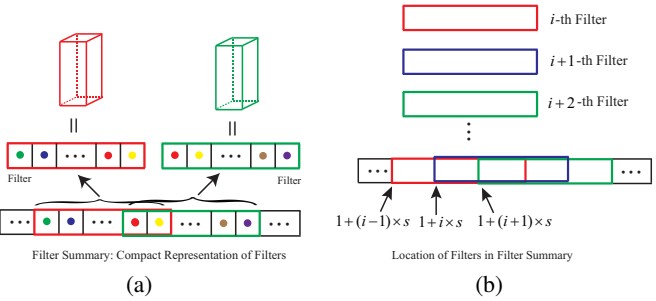

$$1+(i-1)\times s \quad 1+i\times s \quad 1+(i+1)\times s$$

(a) Filter Summary: Compact Representation of Filters     (b) Location of Filters in Filter Summary

Figure 1: (a) Illustration of a Filter Summary (FS). The two filters marked in red and green are two overlapping segments in the FS. The two filters share two weights, indicated as red and yellow dots, in their overlapping region. In this example, the stride for extracting filters is $s = K - 2$ for illustration purpose, since every two neighboring filters share two weights. (b) Illustration of location of filters in the FS. The filter stride is $s = \lfloor \frac{L-1}{C_{\text{out}}} \rfloor$. The $i$-th segment or filter in the FS has $K$ elements with indices $1 + (i-1)s : (i-1)s + K - 1$, for $i \in [C_{\text{out}}]$. In this way, the $C_{\text{out}}$ filters reside in the FS in a weight sharing manner. Padding is used so that the last filter has valid elements.

Formally, let a convolution layer have $C_{\text{out}}$ filters of size $C_{\text{in}} \times S_1 \times S_2 = K$ where $(S_1, S_2)$ is the spatial size and $C_{\text{in}}$ is the channel size. So each filter has $K$ elements, and the total number of parameters of all the filters are $KC_{\text{out}}$. We propose a compact representation of these parameters, which is a 1D vector named Filter Summary (FS), denoted by $\mathbf{F}$. Suppose the goal is to reduce the number of parameters in the convolution layer, i.e. $KC_{\text{out}}$, by $r$ times, then the length of the FS is $L \triangleq \lfloor \frac{KC_{\text{out}}}{r} \rfloor$. In the following text, $r$ is referred to as the compression ratio for convolution.

**For the purpose of the fast convolution algorithm introduced in Section 2.2, each filter is represented as a 1D segment of length $K$ in the FS**. With a defined filter stride $s = \lfloor \frac{L-1}{C_{\text{out}}} \rfloor$ [1], the $i$-th filter can be represented as a segment of length $K$ staring at location $1 + (i - 1)s$ in the FS, i.e. $\mathbf{F}_{1+(i-1)s:1+(i-1)s+K-1}$. With $i$ ranging over $[1 \ldots C_{\text{out}}]$, we have $C_{\text{out}}$ filters residing in the FS in a weight sharing manner. Note that padding is adopted so that the last filter, i.e. the $C_{\text{out}}$-th filter, has valid elements. We ignore such padding for reduced amount of notations in the following text.

Due to weight sharing, the size of FS is much smaller than that of independent $C_{\text{out}}$ filters. It can be seen that FS can be used to compress all the variants of convolution layers in regular CNNs, including convolution layers with filters of spatial size $1 \times 1$. FSNet and its baseline CNN have the same architecture except that each convolution layer of FSNet has a compact representation of filters, namely a FS, rather than a set of independent filters in the baseline. FS is designed such that the number of filters in it is the same as the number of filters in the corresponding convolution layer of the baseline CNN. A more concrete example is given here to describe the compact convolution layer of FSNet. Suppose that a convolution layer of the baseline CNN has 64 filters of channel size 64 and spatial size $3 \times 3$, and the compression ratio for convolution is $r = 4$. Then the corresponding convolution layer in the FSNet has a FS of length $\frac{64 \times 64 \times 3 \times 3}{4} = 9216 = L$. The 64 filters of size $64 \times 3 \times 3 = 576$ are segments located by striding along the FS by $s = \lfloor \frac{9215}{64} \rfloor = 143$ elements.

---

[1]In order to keep notations simple, we let $s = \lfloor \frac{L-1}{C_{\text{out}}} \rfloor$. Please refer to more details in the appendix.

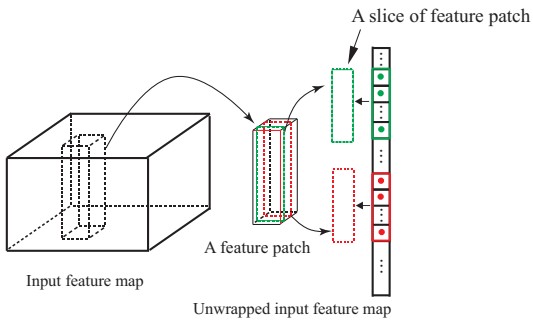

Figure 2: Illustration of unwrapping of the input feature map.

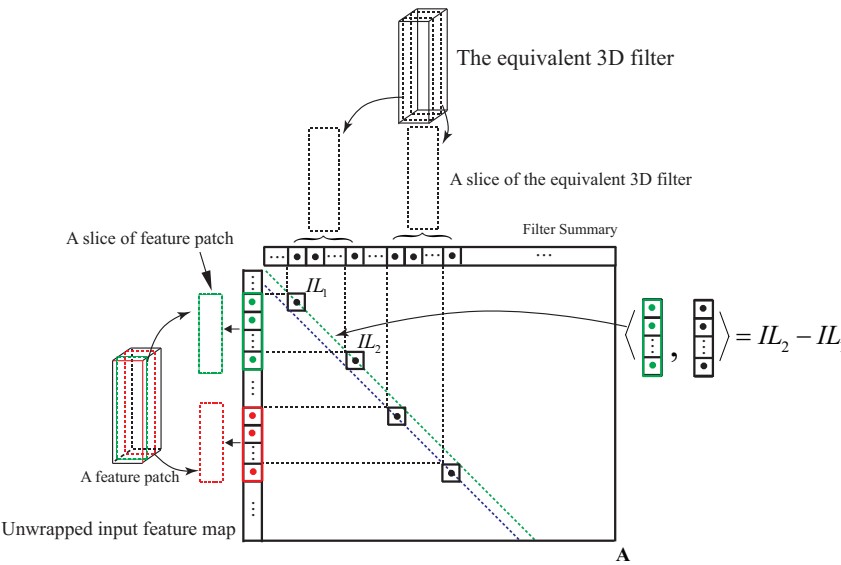

Figure 3: Illustration of efficient convolution using the Filter Summary (FS).

## 2.2 FAST CONVOLUTION BY FILTER SUMMARY (FCFS)

Representing filters as overlapping segments in FS leads to accelerated convolution, which is detailed in this subsection. Convolution computes the inner product between patches of the input feature map and all the filters of a convolution layer. Feature patches usually share weights due to small convolution stride. Therefore, for the case of FS wherein filters also share weights, conventional convolution unnecessarily computes the product between elements of feature patches and elements of the filters many times when these elements of features patches or filters are shared weights.

To handle this problem, we propose Fast Convolution by Filter Summary (FCFS) which avoids the unnecessary computation. The 1D representation of filters facilitates the design of FCFS. We unwrap the input feature map of size $C_{\text{in}} \times D_1 \times D_2$ and reshape it as a 1D vector, denoted by $\mathbf{M}$. Here $(D_1, D_2)$ is the spatial size of the input feature map. The unwrapping of the feature map is in channel-major order, and element with indices $(i, j, k)$ corresponds to $\mathbf{M}_{k \cdot C_{\text{in}} D_1 + j \cdot C_{\text{in}} + i}$. Therefore, different slices of a feature patch of size $C_{\text{in}} \times S_1 \times S_2$ appear in different locations of $\mathbf{M}$. For example, in Figure 2, two slices, marked in green and blue, of a feature patch are two segments of $\mathbf{M}$ where each segment is of size $C_{\text{in}} S_1$.

Let $\mathbf{F}$ denote the FS of length $L$ in the convolution layer. Each filter in the FS can also be viewed as an unwrapped version of an equivalent conventional 3D filter in channel-major order, so different slices of the equivalent 3D filter are different segments of FS, as illustrated by the upper part of Figure 3. To circumvent unnecessary multiple computation of product between weights shared in

feature patches and filters, we compute the product between all the elements of $\mathbf{M}$ and $\mathbf{F}$, and store the result as a matrix $\mathbf{A}$ with $\mathbf{A}_{ij} = \mathbf{M}_i\mathbf{F}_j, i \in [C_{\text{in}}D_1D_2], j \in [L]$. Because each slice of a feature patch or a filter is a segment of $\mathbf{M}$ or $\mathbf{F}$, it can be verified that each line inside $\mathbf{A}$ which is either the principal diagonal of $\mathbf{A}$ or parallel to the principal diagonal contributes to the inner product between slices of feature patch and that of filter. Such lines are defined as the critical lines of $\mathbf{A}$. Two examples of the critical lines are illustrated in Figure 3 by dashed lines in green and blue that cross $\mathbf{A}$, where the dashed line in red is the principal diagonal of $\mathbf{A}$ and the dashed line in blue is parallel to the principal diagonal. For each critical line of $\mathbf{A}$ denoted by a vector $I$ of length $T$, we compute its 1D integral image or integral line denoted by $IL$, where $IL_i = \sum_{t=1}^{i} I_t, i \in [T]$. The integral lines are co-located with their corresponding critical lines. Figure 3 illustrates the integral line for the principal diagonal of $\mathbf{A}$. It can be seen that the inner product between a slice of feature patch and a slice of filter in FS can be computed efficiently by a subtraction along an integral line. For example, Figure 3 illustrates a case where the inner product of a filter in green and a slice of a filter can be computed as $IL_2 - IL_1$, where $IL$ in this example is the integral line for the principal diagonal of $\mathbf{A}$.

Therefore, one can efficiently compute the inner product between a feature patch and a filter by summing the inner products for their $S_2$ pairs of slices, where one pair of slices has a slice of the feature patch and one proper slice of the filter. By concatenating such results for all feature patches and filters, one obtain the final result of the convolution between the input feature map and all the filters in FS in an efficient manner. After the above full description of FCFS, it can be verified that FCFS only needs to compute $\frac{L}{C_{\text{in}}S_1} = L'$ elements for each row of $\mathbf{A}$. This is due to the fact that different slices of filters are separated by multiplies of $C_{\text{in}}S_1$ elements in $\mathbf{F}$. It should be emphasized that FCFS can accelerate convolution only for convolution layer with kernel size $S_2 > 1$. The formal formulation of FCFS is described by Algorithm 1 and Algorithm 2 in Section A.2.

## 2.3 COMPUTATIONAL ACCELERATION RATIO OF FCFS

The computational complexity of conventional convolution for an input feature map of size $C_{\text{in}} \times D_1 \times D_2$ and $C_{\text{out}}$ filters, each of which has a size of $C_{\text{in}} \times S_1 \times S_2 = K$, is $C_{\text{out}}D_1D_2K$. It can be verified that the FCFS algorithm in Section 2.2 requires $2C_{\text{in}}D_1D_2L' + C_{\text{out}}D_1D_2S_2$ steps. Such complexity result can be obtained by noticing that FCFS has three stages. The first stage is the computation of $\mathbf{A}$ which takes $C_{\text{in}}D_1D_2L'$ steps since the size of the portion of $\mathbf{A}$ contributing to convolution result is $C_{\text{in}}D_1D_2L'$. The second stage is the computation of integral lines for all the critical lines of $\mathbf{A}$ which also takes $C_{\text{in}}D_1D_2L'$ steps. The third stage is to compute the inner product between every feature patch and every filter by summing the inner products of their corresponding $S_2$ pairs of slices, and it takes $S_2$ steps for the inner product of a feature patch and a filter by virtue of the integral lines. Concatenating such results form the final result of the convolution. The computational complexity of the third stage is $C_{\text{out}}D_1D_2S_2$. If we only consider the steps involving floating-point multiplication, then the computational complexity of FCFS is $C_{\text{in}}D_1D_2L' + C_{\text{out}}D_1D_2S_2$ since the second stage only has addition/subtraction operations. [2] Therefore, the computational acceleration ratio of FCFS with respect to the conventional convolution is

$$\frac{C_{\text{out}}D_1D_2K}{C_{\text{in}}D_1D_2L' + C_{\text{out}}D_1D_2S_2}.$$

Because $K \gg S_2$ in most cases, the above acceleration ratio is approximately $\frac{C_{\text{out}}D_1D_2K}{C_{\text{in}}D_1D_2L'} \approx rS_1$, which is the product of $S_1$ and the compression ratio for the convolution layer. Note that the depthwise separable convolution in MobileNets (Howard et al., 2017) has computational acceleration ratio of around $S_1 \times S_2$, and FCFS has even faster convolution when the compression ratio $r > S_2$.

## 2.4 TRAINING FSNET

Given a baseline CNN, we design FSNet by replacing each the convolution layer of the baseline CNN by a convolution layer with a FS. The weights of FSNet are trained using regular back-propagation. The detailed training procedure is described in Section 3.

---

[2]This is reasonable since the $C_{\text{out}}D_1D_2K$ steps for the conventional convolution also ignores addition operations.

## 2.5 FSNet with Weight Quantization (FSNet-WQ)

The model size of FSNet can be further compressed by weight quantization. FSNet with Weight Quantization (FSNet-WQ) is proposed to further boost the compression ratio of FSNet, without noticeable loss of prediction accuracy shown in our experiments. The quantization process for FSNet is described as follows. After a FSNet is trained, a one-time linear weight quantization is applied to the obtained FSNet. More concretely, 8-bit linear quantization is applied to the filter summary of each convolution layer and the fully connected layer of the FSNet. 256 levels are evenly set between the maximum and minimum values of the weights of a layer to be quantized, and then each weight is set to its nearest level. In this way, a quantized weight is represented by a byte together with the original maximum and minimum values stored in each quantized layer. The resultant model is named FSNet-WQ. The number of effective parameters of FSNet-WQ is computed by considering a quantized weight as $1/4$ parameter since the storage required for a byte is $1/4$ of that for a floating number in our experiments.

## 2.6 Extension to Differentiable FSNet (DFSNet)

It can be observed that location of filters determines the way how filters share weights in FS. So far the filters are evenly located in FS. It remains an interesting question whether the way filters share weights can be learned, which potentially leads to a better way of weight sharing across the filters. To this end, we propose Differentiable FSNet (DFSNet) in this subsection, where the location of each filter is a learnable parameter and it can be a fraction number rather than the integer location considered so far. Formally, the staring location of a filter $\mathbf{g}$ of length $K$ in a FS $\mathbf{F}$ of length $L$ is parameterized by a parameter $\alpha$ through the sigmoid function as $l = \frac{1}{1+e^{-\alpha}} \cdot L$, and $\mathbf{g}$ can be represented as $\mathbf{g} = (1 + \lfloor l \rfloor - l)\mathbf{F}_{\lfloor l \rfloor : \lfloor l \rfloor + K - 1} + (l - \lfloor l \rfloor)\mathbf{F}_{\lfloor l \rfloor + 1 : \lfloor l \rfloor + K}$. All such $\alpha$ specifying the location of filters in FS are learned during the training of DFSNet in an end-to-end manner. $l$ is almost impossible to be an integer, so the derivative with respect to $\alpha$ almost always exists. Note that one may not use FCFS to accelerate convolution operation in DFSNet due to the fraction location of the filters.

## 3 Experimental Results

We conduct experiments with CNNs for image classification and object detection tasks in this section, demonstrating the compression results of FSNet.

## 3.1 FSNet for Classification

We show the performance of FSNet in this subsection by comparative results between FSNet and its baseline CNN for classification task on the CIFAR-10 dataset (Krizhevsky, 2009). Using ResNet (He et al., 2016) or DenseNet (Huang et al., 2017) as baseline CNNs, we design FSNet by replacing all the convolution layers of ResNet or DenseNet by convolution layers with FS. We train the baseline CNNs using 300 training epochs. The initial learning rate is $0.1$, and it is divided by 10 at $50\%$ and $75\%$ of the 300 epochs. The test accuracy and the parameter number of all the models are reported in Table 1 with compression ratio for convolution $r = 4$. In the sequel, CR stands for compression ratio computed by the ratio of the number of parameters of the baseline CNN over that of the obtained FSNet. It can be observed in Table 1 that FSNet with a compact parameter space achieves accuracy

Table 1: Performance of FSNet on the CIFAR-10 dataset

| Model | | Before Compression | | FSNet | | CR | FSNet-WQ | | CR |
|---|---|---|---|---|---|---|---|---|---|
| | | # Param | Accuracy | # Param | Accuracy | | # Param | Accuracy | |
| ResNet | ResNet-110 | 1.74M | 93.91% | 0.44M | 93.81% | 3.95 | 0.12M | 93.81% | 14.50 |
| | ResNet-164 | 1.73M | 94.39% | 0.47M | 94.59% | 3.68 | 0.16M | 94.65% | 10.81 |
| DenseNet | DenseNet-100 ($k = 12$) | 1.25M | 95.31% | 0.37M | 94.46% | 3.38 | 0.15M | 94.40% | 8.33 |
| | DenseNet-100 ($k = 24$) | 4.83M | 95.71% | 1.33M | 95.40% | 3.63 | 0.44M | 95.43% | 10.98 |

comparable with that of different baselines including ResNet-110 and ResNet-164, DenseNet-100 with growth rate $k = 12$ and $k = 24$. We use the idea of cyclical learning rates (Smith, 2015) for training FSNet. 4 cycles are used for training FSNet, and each cycle uses the same schedule of learning rate and same number of epochs as that of the baseline. A new cycle starts with the initial

learning rate of $0.1$ after the previous cycle ends. The cyclical learning method is only used for FSNet on the CIFAR-10 dataset, and the training procedure of FSNet is exactly the same as that of its baseline throughout all the other experiments.

Table 1 also shows that FSNet with weight quantization, or FSNet-WQ, boosts the compression ratio without sacrificing performance. The effective number of parameters defined in Section 2.5 is reported. FSNet-WQ achieves more than $10\times$ compression ratio for all the two types of ResNet and DenseNet-100 with $k = 24$, and it has less than $0.4\%$ accuracy drop for ResNet-110 and DenseNet-100 with $k = 24$. It is interesting to observe that FSNet-WQ even enjoys slight better accuracy than FSNet for ResNet-164 and DenseNet-100 with $k = 24$. In addition, FSNet-WQ does not hurt the accuracy of FSNet for ResNet-110. We argue that weight quantization imposes regularization on the filter summary which reduces the complexity of the filter summary thus improve its prediction performance. We further demonstrate that FSNet and FSNet-WQ achieve better accuracy and compression ratio with respect to the competing filter pruning method (Li et al., 2017) in Table 9. The model size of FSNet-WQ is less than $1/10$ of that of (Li et al., 2017) while the accuracy is better than that of the latter by $0.5\%$.

Table 2: Comparison between FSNet and the filter pruning method in (Li et al., 2017) on the CIFAR-10 dataset

| Model \ Performance | # Params | Accuracy |
|---|---|---|
| ResNet-110 | 1.74M | 93.91% |
| Filter Pruning (Li et al., 2017) | 1.16M | 93.30% |
| FSNet | 0.44M | 93.81% |
| FSNet-WQ | 0.12M | 93.81% |

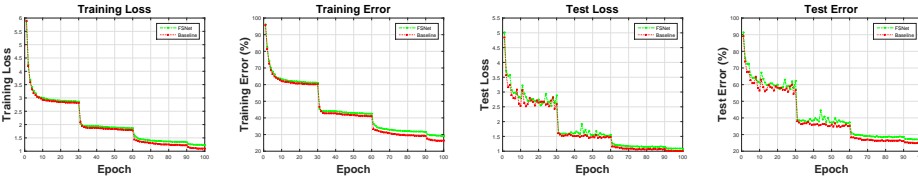

Figure 4: From left to right: training loss, training error, test loss and test error of FSNet and the baseline ResNet-50 on the ImageNet dataset. FSNet has a compression ratio for convolution $r = 2$.

In order to evaluate FSNet on large-scale dataset, Table 3 shows its performance on ILSVRC-12 dataset (Russakovsky et al., 2015) using ResNet-50 as baseline. The accuracy is reported on the standard 50k validation set. We train ResNet-50 and the corresponding FSNet for 100 epochs. The initial learning rate is $0.1$, and it is divided by 10 at epoch $30, 60, 90$ respectively. We tested two settings with compression ratio for convolution being $r = 2$ and $r = 3.7$, and the corresponding two models are denoted by FSNet-1 and FSNet-2 respectively. The training loss/error and test loss/error of FSNet-1 are shown in Figure 4. We can see that the patterns of training and test of FSNet are similar to that of its baseline, ResNet-50, on the ILSVRC-12 dataset. Table 4 shows the comparative results between FSNet-1-WQ and ThiNet (Luo et al., 2017), a competing compression method by filter pruning. We can observe that the quantized FSNet-1 render smaller model size and reduced FLOPs. Here the number of GFLOPs (1 GFLPs = $10^9$ FLOPs) for FSNet and FSNet-WQ are almost the same, and the details are introduced in Section A.3.

## 3.2 FSNET FOR OBJECT DETECTION

We evaluate the performance of FSNet for object detection in this subsection. The baseline neural network is the Single Shot MultiBox Detector (SSD300) (Liu et al., 2016). The baseline is adjusted by adding batch normalization (Ioffe & Szegedy, 2015) layers so that it can be trained from scratch. Both SSD300 and FSNet are trained on the VOC 2007/2012 training datasets, and the mean average precision (mAP) is reported on the VOC 2007 test dataset shown in Table 5. We employ two versions of FSNet with different compression ratios for convolution, denoted by FSNet-1 and FSNet-2 respectively. Again, weight quantization either slightly improves mAP (for FSNet-1), or only slightly hurts it (for FSNet-2). Compared to Tiny SSD (Wong et al., 2018), FSNet-1-WQ enjoys smaller

Table 3: Performance of FSNet on ImageNet

| Model / Performance | # Params | Top-1 | Top-5 |
|---|---|---|---|
| ResNet-50 | 25.61M | 75.11% | 92.61% |
| FSNet-1 | 13.9M | 73.11% | 91.37% |
| FSNet-1-WQ | 3.55M | 72.59% | 91.20% |
| FSNet-2 | 8.49M | 70.36% | 89.79% |
| FSNet-2-WQ | 2.20M | 69.87% | 89.61% |

Table 4: Comparative results of FSNet on ImageNet

| Model / Performance | # Params | Top-1 | GFLOPs |
|---|---|---|---|
| ThiNet (Luo et al., 2017) | 12.38M | 71.01% | 3.41 |
| FSNet-1-WQ | 3.55M | 72.59% | 2.47 |

parameter space while its mAP is much better. Note that while the reported number of parameters of Tiny SSD is 1.13M, its number of effective parameters is only half of this number. i.e. 0.565M, as the parameters are stored in half precision floating-point. In addition, the model size of FSNet-1-WQ is 1.85MB, around 20% smaller than that of Tiny SSD, 2.3MB. It is also interesting to observe that our FSNet-2 and FSNet-2-WQ are both smaller than MobileNetV2 SSD-Lite (Sandler et al., 2018) with better MAP. Since MobileNetV2 SSD-Lite is believed to be newer than MobileNetV1 SSD, the latter is not reported in this experiment.

Table 5: Performance of FSNet for object detection

| Model | # Params | mAP |
|---|---|---|
| SSD300 | 26.32M | 77.31% |
| Tiny SSD (Wong et al., 2018) | 0.56M | 61.3% |
| MobileNetV2 SSD-Lite (Sandler et al., 2018) | 3.46M | 68.60% |
| FSNet-1 | 1.67M | 67.60% |
| FSNet-1-WQ | 0.45M | 67.63% |
| FSNet-2 | 2.59M | 70.14% |
| FSNet-2-WQ | 0.68M | 70.00% |

## 3.3 USING DFSNET FOR NEURAL ARCHITECTURE SEARCH

We also study the performance of integrating FS into Neural Architecture Search (NAS). The goal of NAS is to automatically search for relatively optimal network architecture for the sake of obtaining better performance than that of manually designed neural architecture. We adopt Differentiable Architecture Search (DARTS) (Liu et al., 2019) as our NAS method due to its effective and efficient searching scheme, where the choice for different architectures is encoded as learnable parameters which can be trained in an end-to-end manner. Since DFSNet is also a differentiable framework for model compression, we combine DFSNet and DARTS so as to search for a compact neural architecture aiming at great performance. We design a DFSNet-DARTS model by replacing all the $1 \times 1$ convolution layers, including those for the depthwise separable convolution, in the DARTS search space with FS convolution layers. We perform NAS on the CIFAR-10 dataset using the DFSNet-DARTS model following the training procedure described in DARTS (Liu et al., 2019), and report the test accuracy and model size of the obtained neural network after searching in Table 6. It can be observed that the model found by DFSNet-DARTS has 40% less parameters than that by DARTS while the accuracy loss is only 0.31%, clearly indicating the effectiveness of FS convolution in the task of NAS. Please refer to Section A.4 for more details about DFSNet-DARTS.

Table 6: Performance of DFSNet-DARTS on the CIFAR-10 dataset

| Model | # Params | Accuracy |
|---|---|---|
| DARTS | 3.13M | 97.50% |
| DFSNet-DARTS | 1.88M | 97.19% |

## 4 CONCLUSION

We present a novel method for compression of CNNs through learning weight sharing by Filter Summary (FS). Each convolution layer of the proposed FSNet learns a FS wherein the convolution filters are overlapping 1D segments, and nearby filters share weights naturally in their overlapping regions. By virtue of the weight sharing scheme, FSNet enjoys fast convolution and much smaller parameter space than its baseline while maintaining competitive predication performance. The compression ratio is further improved by one-time weight quantization. Experimental results demonstrate the

effectiveness of FSNet in tasks of image classification and object detection. Differentiable FSNet (DFSNet) is further proposed wherein the location of filters in FS are optimized in a differentiable manner in the end-to-end training process, and the performance of DFSNet is shown by experiment with Neural Architecture Search (NAS).

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

# A APPENDIX

## A.1 ADDITIONAL EXPERIMENTAL RESULT

We demonstrate the performance of FSNet in this subsection by more comparative results between FSNet and its baseline CNN for classification task on the CIFAR-10 dataset. Using ResNet (He et al., 2016) or DenseNet (Huang et al., 2017) as baselines originally designed for ImageNet data, we design FSNet by replacing all the convolution layers of ResNet or DenseNet by convolution layers with FS. We train FSNet and its baseline CNN following the same training procedure specified in the paper, and show the test accuracy and the parameter number of all the models in Table 7.

Table 7: Performance of FSNet on the CIFAR-10 dataset, using larger ResNet and DenseNet as baselines

| Model | | Before Compression | | FSNet | | CR | FSNet-WQ | | CR |
|---|---|---|---|---|---|---|---|---|---|
| | | # Param | Accuracy | # Param | Accuracy | | # Param | Accuracy | |
| ResNet | ResNet-18 | 11.18M | 94.18% | 0.81M | 93.93% | 13.80 | 0.22M | 93.91% | 50.82 |
| | ResNet-34 | 21.30M | 94.72% | 1.68M | 94.29% | 12.68 | 0.45M | 94.32% | 47.33 |
| | ResNet-50 | 23.57M | 95.16% | 2.51M | 94.91% | 9.39 | 0.72M | 94.92% | 32.73 |
| | ResNet-101 | 42.61M | 95.62% | 4.84M | 95.23% | 8.80 | 1.38M | 95.23% | 30.88 |
| DenseNet | DenseNet-121 | 7.04M | 95.13% | 1.24M | 95.11% | 5.68 | 0.44M | 95.13% | 16 |

Because the results of FSNet in Table 1 are obtained by cyclic training with four cycles, we show the performance of FSNet using the same training procedure (with only one training cycle) as its baseline in Table 8. We can see that the accuracy loss of FSNet is still less than 1% compared to its baseline, and FSNet with one cycle has even higher accuracy (95.46%) than that with four cycles (95.40%) for DenseNet-100 with a growth rate of 24.

Furthermore, we show the performance of FSNet with ResNet-56 on the CIFAR-10 dataset with different compression ratios, and compare FSNet to Discrimination-aware Channel Pruning (DCP) method (Zhuang et al., 2018). It can be observed that with the compression ratio of 4.22 and 4.58, FSNet-1 and FSNet-2 achieve the accuracy of 94.08% and 94.12%. Both of them enjoy smaller models with higher accuracy compared to DCP (Zhuang et al., 2018). In addition, a very aggressive compression ratio of 13.76 pushes the accuracy down to 92.87%, and the incurred accuracy loss is still less than 1% compared to the original ResNet-56.

Table 8: Performance of FSNet with one-cycle (the same training procedure as the baseline) on the CIFAR-10 dataset

| Model | | Before Compression | | FSNet | | CR | FSNet-WQ | | CR |
|---|---|---|---|---|---|---|---|---|---|
| | | # Param | Accuracy | # Param | Accuracy | | # Param | Accuracy | |
| ResNet | ResNet-110 | 1.74M | 93.91% | 0.44M | 93.02% | 3.95 | 0.12M | 93.03% | 14.50 |
| | ResNet-164 | 1.73M | 94.39% | 0.47M | 93.75% | 3.68 | 0.16M | 93.84% | 10.81 |
| DenseNet | DenseNet-100 ($k = 12$) | 1.25M | 95.31% | 0.37M | 94.36% | 3.38 | 0.15M | 94.31% | 8.33 |
| | DenseNet-100 ($k = 24$) | 4.83M | 95.71% | 1.33M | 95.46% | 3.63 | 0.44M | 95.47% | 10.98 |

Table 9: Comparison between FSNet and the Discrimination-aware Channel Pruning (DCP) method (Zhuang et al., 2018) on the CIFAR-10 dataset, where the $\times \downarrow$ shows the compression ratio (the ratio of the number of parameters of the original model to that of the compressed model).

| Model Performance | # Params | Accuracy |
|---|---|---|
| ResNet-56 | 0.86M | 93.80% |
| DCP (Zhuang et al., 2018) | 3.37× ↓ | 93.81% |
| FSNet-1 | 4.22× ↓ | 94.08% |
| FSNet-2 | 4.58× ↓ | **94.12%** |
| FSNet-3 | 4.94× ↓ | 93.74% |
| FSNet-4 | 13.76× ↓ | 92.87% |

Table 10: Performance of FSNet-WQ and Linear Quantization on CIFAR-10

| Model | # Params | Accuracy |
|---|---|---|
| ResNet-164 | 1.73M | 94.39% |
| FSNet-WQ | 0.16M | 94.64% |
| Linear Quantization | 0.50M | 94.36% |

Table 11: Performance of FSNet-WQ and Linear Quantization on ImageNet

| Model | # Params | Top-1 | Top-5 |
|---|---|---|---|
| ResNet-50 | 25.61M | 75.11% | 92.61% |
| FSNet-WQ | 3.55M | 72.59% | 91.20% |
| Linear Quantization (8-bit) | 8.56M | 74.88% | 92.48% |
| Linear Quantization (4-bit) | 5.71M | 0.086% | 0.578% |

We are also interested in the visualization of the learned Filter Summary. For the visualization purpose, we change the representation of a Filter Summary from a 1D vector to a 3D tensor. We design a 3D Filter Summary (FS) for the first convolution layer (conv1) of Network In Network (NIN) (Lin et al., 2014) on the CIFAR-10 dataset. The conv1 of the original NIN has 192 filters of spatial size $5 \times 5$ and channel size 3. The 3D Filter Summary for conv1 of the FSNet version of NIN is a 3D tensor of size $24 \times 8 \times 3$, with a spatial size of $24 \times 8$ and a channel size of 3. In the forward process, 192 filters of size $5 \times 5 \times 3$ are extracted as overlapping 3D tensors from the 3D FS. The illustration of the learned 3D FS for conv1 of the FSNet version of NIN and the independent 192 filters learned in conv1 of the original NIN are shown in Figure 5. We can observe that, compared to the independent 192 filters, the filters in the 3D FS are "smoothed" by the weight sharing scheme because patterns of one filter are shared across its neighboring filters in the 3D FS.

## A.2 MORE DETAIL ABOUT FILTER STRIDE $s$ IN FAST CONVOLUTION BY FILTER SUMMARY (FCFS)

In the paper, it is mentioned that the filter stride is set to $s = \lfloor \frac{L-1}{C_{\text{out}}} \rfloor$. In order to accelerate the convolution operation using FS for convolution layer with $S_2 > 1$, we require that $s$ be the largest multiple of $C_{\text{in}} S_1$ which is no greater than $\lfloor \frac{L-1}{C_{\text{out}}} \rfloor$. Namely, let $s_1 = \lfloor \frac{L-1}{C_{\text{out}}} \rfloor$, then $s = \lfloor \frac{s_1}{C_{\text{in}} S_1} \rfloor \cdot C_{\text{in}} S_1$. The reason is that FS can be viewed as the unwrapped version of a conventional 3D filter by shaping its elements into a vector in the order of channel, the first spatial dimension (with size $S_1$, the second spatial dimension (with size $S_2$). Therefore, different slices along the second spatial dimension are separated by $C_{\text{in}} S_1$ elements in the FS. By letting $s$ be the multiple of $C_{\text{in}} S_1$, we only need to compute $\frac{L}{C_{\text{in}} S_1} = L'$ elements for each row of matrix $\mathbf{A}$.

Algorithm 1 describes the algorithm which computes the matrix $\mathbf{A}$ for FCFS. Algorithm 2 describes the FCFS algorithm which computes the convolution between the $i$-th filter in the FS of a convolution layer of FSNet and the input feature map $\mathbf{M}$ of size $C_{\text{in}} D_1 D_2 \times 1$ for that convolution layer. The output feature map for that convolution layer of FSNet is obtained by running Algorithm 2 for all $i \in [C_{\text{out}}]$. Throughout this paper, we let the convolution stride be 1. FCFS can be be used to accelerate convolution with different convolution strides.

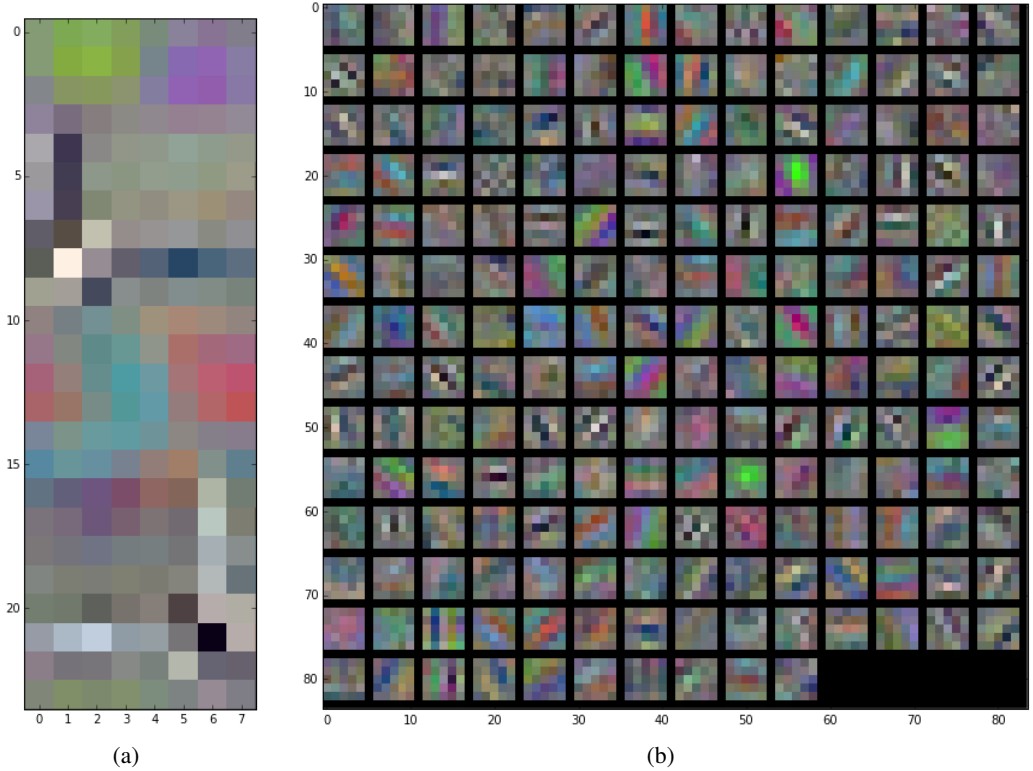

(a)            (b)

Figure 5: (a) Illustration of the learned 3D Filter Summary (FS) for the first convolution layer of Network In Network (NIN) (Lin et al., 2014), where 192 filters of size $5 \times 5 \times 3$ are extracted as overlapping 3D tensors from the 3D FS in the forward process. (b) The learned 192 independent filters in the first convolution layer of the original NIN.

---

**Algorithm 1** Compute the matrix $\mathbf{A}$ for FCFS

---

**Input:**
    The filter summary $\mathbf{F}$ of size $L = C_{\text{out}}K$ in a convolution layer of FSNet, the input feature map $\mathbf{M}$ of size $C_{\text{in}}D_1D_2 \times 1$.
1: **for** $1 \leq t \leq C_{\text{in}}D_1D_2$ **do**
2:     $j = t \mod C_{\text{in}}$
3:     **while** $j \leq L$ **do**
4:         $\mathbf{A}_{ij} = \mathbf{M}_i\mathbf{F}_j$
5:         $j = j + C_{\text{in}}S_1$
6:     **end while**
7: **end for**
8: **for** Each critical line $\mathbf{I}$ of $\mathbf{A}$ **do**
9:     Compute the 1D integral image $IL$ of $\mathbf{I}$, where $IL_i = \sum_{t=1}^{i} \mathbf{I}_t, i \in [T]$ and $T$ is the length of $\mathbf{I}$
10:     Assign $IL_i$ to the co-located element of $\mathbf{A}$
11: **end for**
**Output:**   The matrix $\mathbf{A}$

---

### A.3 MORE DETAILS ABOUT FLOPS OF FSNET-WQ

It is mentioned that FSNet-WQ has almost the same number of FLOPs as that of FSNet. The underlying reason is explained in detail in this section. Suppose a convolution layer or fully-connected layer of FSNet performs the following operation:

$$\mathbf{y} = \mathbf{W}\mathbf{x} + \mathbf{b}, \tag{1}$$

where $\mathbf{x}$ is a vector representing the input of this layer, $\mathbf{y}$ is the output, and the operation performed by this layer is represented by a linear function parameterized by $\mathbf{W}$ and $\mathbf{b}$. Let $\mathbf{W}_0$ and $\mathbf{b}_0$ denote the quantized $\mathbf{W}$ and

---

**Algorithm 2** Convolution between the $i$-th filter in the filter summary and the input feature map for a convolution layer of FSNet

---

**Input:**

The filter summary $\mathbf{F}$ of size $L = C_{\text{out}}K$ in a convolution layer of FSNet, the filter index $i \in [C_{\text{out}}]$, the input feature map $\mathbf{M}$ of size $C_{\text{in}}D_1D_2 \times 1$, the matrix $\mathbf{A}$ computed by Algorithm 1

1: Build a matrix $\mathbf{O}$ of size $D_1 \times D_2$ with zero initialization
2: Obtain the padded input feature map of size $C_{\text{in}}(D_1 + S_1 - 1)(D_2 + S_2 - 1) \times 1$, which is still denote by $\mathbf{M}$
3: **for** $1 \leq m \leq D_1$ **do**
4:   **for** $1 \leq n \leq D_2$ **do**
5:     **for** $1 \leq k \leq S_2$ **do**
6:       $t = (n + k - 2)C_{\text{in}}D_1 + (m - 1)C_{\text{in}}$
7:       Obtain the inner product $p_k$ between the slice of the input feature patch $\mathbf{M}_{t:t+C_{\text{in}}S_1}$ and the corresponding slice of the $i$-th filter, i.e. $\mathbf{F}_{1+(i-1)s+(k-1)C_{\text{in}}S_1:(i-1)s+kC_{\text{in}}S_1}$ using the corresponding critical line of $\mathbf{A}$
8:       $\mathbf{O}_{mn} = \mathbf{O}_{mn} + p_k$
9:     **end for**
10:   **end for**
11: **end for**
**Output:** Return $\mathbf{O}$ as the $i$-th channel of the output feature map of the convolution layer of FSNet

---

$\mathbf{b}$ after performing the 256-level linear quantization described in Section 2.5 of the paper. Then the recovered weights from the quantized weights are

$$\tilde{\mathbf{W}} = \mathbf{S}_0 + \tau\mathbf{W}_0, \tilde{\mathbf{b}} = \mathbf{s}_0 + \tau\mathbf{b}_0, \tag{2}$$

where $\mathbf{S}_0$ is a matrix of the same size as $\mathbf{W}$ with all elements being $s_0$ which is the smallest element of $\mathbf{W}$, $\mathbf{s}_0$ is a vector of the same size as $\mathbf{b}$ will all elements being $s_0$. $\tau$ is the quantization step computed by $\tau = \frac{s_1 - s_0}{255}$ and $s_1$ is the maximum element of $\mathbf{W}$. It can be verified that FSNet-WQ performs the following operation in this layer using the recovered weights $\tilde{\mathbf{W}}$ and $\tilde{\mathbf{b}}$:

$$\tilde{\mathbf{y}} = \tilde{\mathbf{W}}\mathbf{x} + \tilde{\mathbf{b}} = \mathbf{S}_0\mathbf{x} + \mathbf{s}_0 + \tau(\mathbf{W}_0\mathbf{x} + \mathbf{b}_0). \tag{3}$$

We can perform the operation $\mathbf{y}_0 = \mathbf{W}_0\mathbf{x} + \mathbf{b}_0$ using the same number of FLOPs as that required for $\mathbf{W}\mathbf{x} + \mathbf{b}$. Then each element of the output of this layer can be computed by $\tilde{\mathbf{y}}_i = (\mathbf{y}_0)_i + s_0(sx + 1)$ where $sx$ is the sum of elements of the input $\mathbf{x}$. Therefore, compared to FSNet, FSNet-WQ only needs additional FLOPs to compute the sum of elements of the input (plus one more multiplication) for each convolution layer or fully-connected layer. Our empirical study shows that the FLOPs of FSNet-WQ and FSNet are almost the same.

After a FSNet is trained, a one-time linear weight quantization is applied to the obtained FSNet. More concretely, 8-bit linear quantization is applied to the filter summary of each convolution layer and the fully connected layer of the FSNet. More comparative results between FSNet-WQ and linear quantization on the CIFAR-10 data set and the ILSVRC-12 dataset are shown in Table 10 and Table 11 respectively. Because linear quantization is used in FSNet-WQ, the compression results of FSNet-WQ are compared to that of linear quantization.

## A.4 MORE DETAILS ABOUT DFSNET-DARTS

We use SGD with momentum of 0.9 to optimize the weights of the DFSNet-DARTS model on the CIFAR-10 dataset. The initial learning rate is 0.01, and the learning rate is gradually reduced to zero following a cosine schedule. The weight decay is set to 0.0002. The DFSNet-DARTS model is trained for 600 epochs with a mini-batch size of 96. The neural architecture found by DFSNet-DARTS is illustrated by Figure 6.

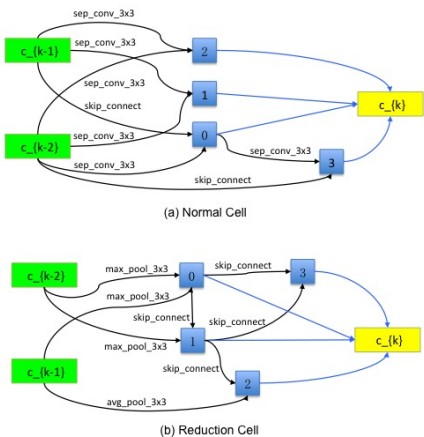

Figure 6: The neural architecture found by DFSNet-DARTS

