# OpenReview forum: "FSNet: Compression of Deep Convolutional Neural Networks by Filter Summary"
_ICLR.cc/2020/Conference — Accept (Poster)_

### Official Review · AnonReviewer1 · 2019-10-13
**Official Blind Review #1**

**Rating:** 6

**Review:**

The paper presents a novel compact parameterization of convolution filters. Normally a filter bank is stored as a 3D array of shape C_out x C_in x H x W = K. This paper proposes to use a flat array instead, and obtain each of the C_out filters of shape C_in x H x W by indexing this array with stride. Thus, adjacent filters will share some of their weights. A fast algorithm is presented for convolving using filters parameterized in this way. An extension is presented where the location of each filter in the flat array is learned. The results show the potential of the method to reduce the number of parameters at a modest drop in accuracy, though it is not clear how the method stacks up against state of the art, or what the improvement in wall-clock runtime is. Overall, I would rate this paper "weak accept".

One minor weakness of the experiments on classification (3.1) is that the training procedure for FSNet and the baselines are different, with FSNet using cyclic learning rates and a larger number of epochs, making the results somewhat difficult to interpret. I suppose that the authors tried using the same training procedure but it did not work for FSNet - is that correct? If not, reporting those numbers would be preferable. If indeed the same training procedure does not work for FSNet that is not a fatal flaw, but we would want to see results for the baseline architectures trained in the same way as FSNet (to make sure the difference is not attributable to the training procedure). Also, we would want to make sure that the baseline results as reported are about as good as they can get for that architecture, e.g. by comparing to results for those architectures published by others, and by copying their training procedure. (I think the baseline results are fine, but it would be good to take away any doubt in the reader's mind).

The paper has a clear structure and is fairly well written, though it may still be beneficial to go over the text with a native speaker.

Overall, I find that the experiments convincingly show that FSNet provides a way to compactly store filters with modest loss of accuracy. I am not an expert in this field though, and so I don't know how it relates to state of the art. One paper that may be good to compare to is "Discrimination-aware Channel Pruning for Deep Neural Networks" by Zhuang et al.

It would be nice to see wall-clock time / speed improvements reported, instead of only reporting the reduction in parameter count.

Typos & minor issues
The paper uses \cite{} in many places where \citep{} would be more appropriate.
"fist spatial size"
"each filer in a"
"substraction"
"epoches"
"efficiency searching scheme"


**Experience Assessment:**

I do not know much about this area.

**Review Assessment: Checking Correctness Of Derivations And Theory:**

I assessed the sensibility of the derivations and theory.

**Review Assessment: Checking Correctness Of Experiments:**

I assessed the sensibility of the experiments.

**Review Assessment: Thoroughness In Paper Reading:**

I made a quick assessment of this paper.

---

> ### Author Response · Authors · 2019-11-15
> **Thank you for your comment**
>
> Thank you for your comment!
>
> 1. “I don't know how it relates to state of the art.”
>
> Our FSNet for object detection, namely FSNet-2-WQ in Section 3.2 of the paper and Table 5, has a model with 0.68M parameters (2.8MB in size) with a mAP of 70.00% on the VOC2007 dataset. This is the smallest model in terms of the number of parameters on this dataset compared to the state-of-the-art models the VOC2007 dataset, including the most recent YOLO Nano [1] (as mentioned by Reviewer #2) with a size of 4MB and a mAP of 69.1% on the same VOC 2007 dataset, and another state-of-the art, MobileNetV2 SSD-Lite (Sandler et al. 2018) with 3.46M parameters and a mAP of 68.60% on the same dataset.
> We have performed comparison between FSnet and Discrimination-aware Channel Pruning [1] by Zhuang et al. on the CIFAR-10 dataset in Table 9 of the revised paper.
>
> 2. "One minor weakness of the experiments on classification (3.1) is that the training procedure for FSNet and the baselines are different."
>
> The only experiment we use different training procedure for FSNet is for the experiment on the CIFAR-10 dataset shown in Table 1, and the training procedure of FSNet is exactly the same as that for its baseline for all the other experimental results.
> In Table 8 of the revised paper, we show the performance of FSNet using the same training procedure (with only one training cycle) as its baseline. We can see that the accuracy loss of FSNet with one cycle is still less than 1% compared to its baseline, and FSNet with one cycle has even higher accuracy (95.46%) than that with four cycles (95.40%) for DenseNet-100 with a growth rate of 24.
>
> 3. "It would be nice to see wall-clock time/speed improvements reported, instead of only reporting the reduction in parameter count."
>
> Thank you for your suggestion! We have demonstrated the speed improvement in terms of the number of Floating Point Operations (FLOPs)  for ResNet-50 on the  ILSVRC- 12 dataset in Table 4 of this paper. Our FSNet-1-WQ in Table 4 with FSNet has 2.47 GFLOPs (1 GFLOPs = 10^9 FLOPs), which is much smaller than that of ThiNet (3.41 GFLOPs). GFLOPs is popularly used for measuring the computation required by a model, and the actual wall-clock inference time is architecture-dependent. However, we will evaluate the actual wall-clock inference time of FSNet.
>
> The typos and grammatical errors have been fixed in the revised paper.
>
>
> [1] Zhuang et al. Discrimination-aware channel pruning for deep neural networks. NeurIPS 2018.
> [2] Wong et al. YOLO Nano: a Highly Compact You Only Look Once Convolutional Neural Network for Object Detection. arXiv: 1910.01271

---

### Official Review · AnonReviewer2 · 2019-10-23
**Official Blind Review #2**

**Rating:** 6

**Review:**

This paper proposes a method, termed as Filter Summary (FS), for weight sharing across filters of each convolution layer. Moreover, a fast convolution algorithm is designed for the convolution layer with the FS. Some promising results demonstrate the effectiveness in CNN compression and the acceleration on the tasks of image classification, object detection and neural architecture search. I like the idea of weight sharing which seems like a reasonable choice for model compression. I expect it to be a general and standard component for model compression and acceleration. I have the following concerns:

1.FSNet achieves a small model (FSNet-2-WQ in Table 5) with only 0.68M parameters and an mAP of 70.00% on the VOC2007 dataset. Can the authors include the model size of FSNet-2-WQ in MB so that the model size can be directly compared to that of the recently proposed YOLO Nano [1] (with a size of 4MB and an mAP of 69.1% on VOC 2007 dataset)?

2. The backbone of FSNet-2-WQ in Table 5 is SSD300, which is a relatively large model for object detection. Please shed more insights on why weight sharing by Filter Summary can have such a high compression ratio, even with the help of quantization.

3 Please explain why the simple linear quantization is compatible with Filter Summary for image classification and object detection. More concretely, why the simple linear quantization does not hurt the performance of FSNet on these tasks? Can similar results be extended to more models and tasks?

4 Please show the illustration of the learned Filter Summary which can help understand the structure of the filters in this new representation.

5 It could be better if the paper can reveal the correlation between the compression ratio and the accuracy of the compressed models by FSNet.

[1] Wong et al. YOLO Nano: a Highly Compact You Only Look Once Convolutional Neural Network for Object Detection. arXiv: 1910.01271


**Experience Assessment:**

I have read many papers in this area.

**Review Assessment: Checking Correctness Of Derivations And Theory:**

I assessed the sensibility of the derivations and theory.

**Review Assessment: Checking Correctness Of Experiments:**

I assessed the sensibility of the experiments.

**Review Assessment: Thoroughness In Paper Reading:**

I read the paper at least twice and used my best judgement in assessing the paper.

---

> ### Author Response · Authors · 2019-11-15
> **Thank you for your comment**
>
> Thank you for your comment!
>
> 1. FSNet-2-WQ has a size of 2.8MB on disk. Therefore, our FSNet-2-WQ has a smaller size and higher mAP (70.00%) compared to YOLO Nano [1] with 4MB size and a mAP of 69.1%.
>
> 2. Our FSNet exploits the redundancy inside the convolutional filters in CNNs by encouraging the weights of filters to share value. This process reduces the complexity of the usually over-parameterized CNNs and remove noise in the filters, so that the prediction accuracy of the original model can still be maintained (or with mild accuracy loss) after significantly reducing the number of parameters.
>
> 3. Linear quantization can be viewed as a regularization on the learned Filter Summary (FS) in FSNet. Note that every element of the FS is shared by multiple filters, so that each element of the FS can be interpreted as a center or mean after applying clustering methods on the traditionally learned filters, so the elements of the FS are robust to quantization, and quantization can play a role of removing unnecessary complexity or noise in the FS so as to improve its generalization and representation capability. We also applied linear quantization to the DFSNet-DARTS model in Table 6 of this paper, and DFSNet-DARTS with linear quantization only has 0.47M parameters with accuracy still higher than 97% on the CIFAR-10 dataset.
>
> 4. Such illustration for ResNet-101 on the CIFAR-10 dataset has shown in Figure 5 in the revised paper.
>
> 5. In Table 3, we demonstrate the performance of FSNet with two different compression ratios, namely FSNet-1 and FSNet-2 with compression ratio being 2 and 3.7 respectively. In Table 5, the performance of FSNet with two different ratios are demonstrated, i.e. FSNet-1 and FSNet-2 with compression ratio being 15.7 and 10.1 respectively.
>
> We conduct more experiments with FSNet for ResNet-56 on the CIFAR-10 datasets with multiple compression ratios for compression, and the relationship between the compression ratio and the classification accuracy on the CIFAR-10 dataset is shown in Table 9 of the revised paper.
>
> [1] Wong et al. YOLO Nano: a Highly Compact You Only Look Once Convolutional Neural Network for Object Detection. arXiv: 1910.01271

---

### Official Review · AnonReviewer4 · 2019-12-04
**Official Blind Review #4**

**Rating:** 8

**Review:**

In this paper authors propose a novel idea (called Filter Summary, FS)  how to compress convolutional neural networks with 2D convolutions (kernels are 3D tensors, it is also applicable to the 1D convolutions). Compression of the convolution operation is done with weight sharing: unwrapped kernel (channel-major) is packed into 1D vector by having intersected segments (shared weights).

The strong sides of the paper are the following:
- Proposed algorithm for efficient computation of convolution operation with FS (less number of multiplication operations with comparison to the standard convolution).
- Via experiments it is demonstrated that proposed approach provides compression of the model while having close to the baseline quality for image classification and object detection tasks and for small and large datasets.
- Proposed method gives better compression factor and better quality than state-ot-the-art models for classification and object detection tasks.
- It is experimentally showed that architecture search works for proposed convolution.

There are still ways to improve the paper:
- In the introduction section it is written that "FS is quantization friendly". I would say the main concern should be "FS quantization achieves the same quality as original FS while having higher compression factor". Standard convolutions are also quantization friendly, no? At the same time I don't see huge gain to do quantization of the model. Yes, to store it we need less memory, however, for the inference we still use float-point computation. How will this quantization be helpful for inference on CPU/GPU (memory/speed)?
- Did you implement CUDA kernel to have efficient FS convolution (I suspect the answer yes)? It would be interesting to see the training time per epoch/forward/backward comparison for standard CNN and FS CNN as soon as you are using the same training process (for models you trained to have an idea for practical usage). Also it is interesting to see comparison of inference time on CPU/GPU.
- For illustrations (figs 1, 2, 3) of input and filters packing it would be very helpful to mark where are channel and spatial dimensions on the figure for simpler understanding the packing. Typos for the figures: "in Figure 2, two slices, marked in green and blue" -> "in Figure 2, two slices, marked in green and red"; "Figure 3 by dashed lines in green and blue that cross A, where the dashed line in red" -> "Figure 3 by dashed lines in green and blue that cross A, where the dashed line in green" (on Figure 3 there are blue and green lines only).
- In the paragraph where the number of elements of matrix A is specified to be computed, it would be better to have detailed explanation why we skip C_in*S_1 elements. It is not so trivial when you read the paper the first time.
- In section 2.3 for complexity of the three stages: seems the third stage doesn't have float-point computations (this is addition for S_2 locations in matrix A), however, the third stage is included into analysis of multiplication operations (this doesn't affect the final result of complexity)
- For DFSNet are filters ordered for FS F (monotonically located)? Or now they can be located independently and starting point defined by its alpha?
- Typo: page 7 "in Table 9" -> "in Table 2"
- In Table 4 it would be better to have results for FSNet-1 too (without quantization).
- In section 3.3 there is no details how the alpha parameter from DFSNet is used in DARTS.
- In Tables 10 and 11 there are linear quantization, what does it mean? what is the difference in quantization procedure with FSNet-WQ (maybe I missed something in the paper).

**Experience Assessment:**

I have read many papers in this area.

**Review Assessment: Checking Correctness Of Derivations And Theory:**

I carefully checked the derivations and theory.

**Review Assessment: Checking Correctness Of Experiments:**

I carefully checked the experiments.

**Review Assessment: Thoroughness In Paper Reading:**

I read the paper thoroughly.

---

### Decision · Program_Chairs · 2019-12-19

**Decision:**

Accept (Poster)

**Comment:**

The paper proposes to compress convolutional neural networks via weight sharing across filters of each convolution layer. A fast convolution algorithm is also designed for the convolution layer with this approach. Experimental results show (i) effectiveness in CNN compression, (ii) acceleration on the tasks of image classification, object detection and neural architecture search. While the authors addressed most of reviewers' concerns, the weakness of the paper which remains is that no wall-clock runtime numbers (only FLOPS) are reported - so efficiency of the approach in practice in uncertain.